# Health Risk Assessment of Trace Metals in Bottled Water Purchased from Various Retail Stores in Pretoria, South Africa

**DOI:** 10.3390/ijerph192215131

**Published:** 2022-11-16

**Authors:** Joshua Oluwole Olowoyo, Unathi Chiliza, Callies Selala, Linda Macheka

**Affiliations:** 1Department of Health Sciences, Marieb College of Health and Human Services, Florida Gulf Coast University, Fort Myers, FL 33965, USA; 2Department of Biology and Environmental Sciences, Sefako Makgatho Health Sciences University, Pretoria P.O. Box 139, South Africa

**Keywords:** trace metals, toxicity, bottled water, health effects, anthropogenic activities

## Abstract

Bottled water is one of the fastest growing commercial products in both developing and developed countries owing to the believe that it is safe and pure. In South Africa, over the years, there has been an increase in the sale of bottled water due to the perceived notion that water supplied by the government may not be safe for human consumption. This study investigated the concentrations of trace metals and the physicochemical properties of bottled water purchased from various supermarkets (registered and unregistered) in Pretoria with a view to determining the health risk that may be associated with the levels of trace metals resulting from the consumption of the bottled water. Twelve commonly available different brands of bottled water were purchased and analysed for trace-metal content using inductively coupled plasma mass spectrometry (ICP-MS). The water samples were also analysed for various physicochemical parameters. The health risk was assessed using the target hazard quotient (THQ). For all the bottled water, the highest concentration of all the elements was recorded for Fe. The values reported for Cr, Ni and Pb were above the limit recommended by World Health Organization. The pH values ranged from 4.67 to 7.26. Three of the samples had pH values in the acidic region below the permissible standard of 6.8–8.0 set by the International Bottled Water Association (IBWA). The target hazard quotient calculated for the water samples showed a minimum risk for Pb, Cr and Ni. The study showed the need to adhere to a strict compliance standard considering the fact that South Africa has rich natural mineral elements, which may have played a role in the high levels of trace metals reported from some of the water samples.

## 1. Introduction

The quality of water varies from source to source and is mostly influenced by natural and anthropogenic factors [1]. Despite the uncertainty surrounding the quality of drinking water, the commercial market for bottled water is increasing globally. Consumer safety is lacking in the verification of chemical-compound contents due to the possible migration of pollutants into the water or from soil within the aquifer and this still remains a cause for concern [2,3,4,5]. According to Gerassimidou et al. [5], in certain temperatures, chemicals have the potential to migrate from polyethylene terephthalate (PET) plastic bottles into the water as well as during recycling. It is also possible for the water to be contaminated from the source due to variations in geological and geographical factors [1].

Globally, freshwater quality is under constant deterioration due to anthropogenic activities such as mining effluent and accidental spills which may affect water quality [6,7,8]. Freshwater systems in most of the developing countries experience pressure from harmful chemicals due to the inability of municipal water-treatment facilities to remove these chemicals or other impurities completely from the water because of the cost implications and old technology as well the increase in demand for portable water supplies [9,10]. Factors such as seasonal changes, climate change, anthropogenic activity and natural phenomena may also affect the chemical composition of water [11].

Toxic trace metals as pollutants may originate from both natural and anthropogenic processes, such as mineral weathering, industrial activities, municipal wastewater discharge, unsustainable agricultural practices and traffic activities [12]. One of the entry points for pollutants into the water-supply system is through leaks and pipeline cross-connections resulting in deterioration of drinking water quality [13]. It is also possible to have metal build-up in the groundwater because of weak bonding to the aquifer especially in the case of boron and chromium as they can also form stable oxy-anions that are not adsorbed by soil, thereby remaining in the water [14]. It should also be noted that various human activities, especially industrial activities, waste disposal and urbanization have affected the original state of natural water [15]. This has evoked awareness and necessitated the need to control the purity of water and to make assessments of its suitability for drinking [16]. Pollutants in drinking water pose serious health risks for infants, the elderly and people with weakened immune systems due to such things as viral infections, immune disorders and cancer [3].

Scientific evidence has shown that bottled water is preferred over municipal drinking tap water due to its taste and perceived safety [17,18]. There are also safety concerns about tap-water plumbing systems and the trustworthiness of municipal governance as compared to bottled water [18,19,20,21]. For instance, the study of Ab Razak et al. [22] showed that corrosion may occur over time leading to release of toxic elements into the water when there is a contact between water and the metal coating of the pipes. However, according to studies by Addisie [21] and Denantes and Donoso [23], the consumer’s risk perception is solely dependent on the taste, odour and clarity of the water. Therefore, bottled water has proliferated the market structure of water consumption in urban areas [2].

Plastic bottles, on the other hand, may cause the release of phthalates and other pollutants which may contribute to the water contamination [5,24]. Shotyk et al. [25] noted that, as a result of different manufacturing processes, water from plastic bottles may become contaminated by Pb and Sb due to the leaching of these elements from the plastic containers used. Prolonged bottled-water storage at high temperatures (50 °C) may also pose a threat to the quality of the water and become a health hazard [26]. Gerassimidou et al. [5] also reported that various food and drink beverages are potentially vulnerable to polyethylene terephthalate (PET) chemicals due to their processing, packaging and transportation.

Water is a vital component of cells, tissues and body organs and is essential for life. It makes up approximately 75% of body weight providing a mineral balance to the human body structure [27,28]. However, considering the fact that South Africa has a rich variety of minerals, with high reserves of iron ore, platinum, manganese, chromium, copper, uranium, silver, beryllium, and titanium among others, the natural water might have been exposed to any of these minerals at levels higher than what is acceptable for human consumption. It is therefore prudent to examine if these naturally occurring minerals have polluted the underground water in its natural state [29]. It should also be noted that South Africa has a significant number of mining industries contributing to its GDP and the processes involved in the excavation of these mineral resources may also pollute the groundwater [30]. Reports from previous studies conducted by Olowoyo et al. [31] showed that there may be elevated concentrations of some toxic trace metals such as Pb, As and Mn in soils of Pretoria, emanating not only from the anthropogenic activities of the area but also from the background levels. Studies from Herselman [32] showed very high concentrations of some elements occurring naturally in soils from South Africa.

High concentrations of trace metals in drinking water may pose a serious human health risk and cause other related health diseases [33]. The World Health Organization (WHO) has set a permissible limit for these trace elements in water and other food substances and concentrations above these limits may cause health problems owing to the levels of consumption and other environmental factors [34]. Generally, the concentration of mineral elements in freshwater is influenced by the type and composition of soil and bedrock through which water flows and there may be elevated concentrations in groundwater with a longer contact period [35]. Anthropogenic activities, on the other hand, play a significant role in the increase of trace elements in the aquatic environment [34]. The current study aimed to assess the quality of commonly consumed bottled water in Pretoria purchased from both registered and unregistered stores with a view to determining the presence of toxic trace metals and the overall possible effect on human health.

## 2. Materials and Methods

### 2.1. Sample Collection

Twelve commonly consumed bottled water samples of different brands were purchased from various registered and unregistered stores in Pretoria, South Africa. The stores carried various brands of bottled water including their own store brands. The labels on each of these brands were checked to determine whether the source of water was “natural spring water”, “prepared water” or “mineral water”. The bottled waters were in plastic bottles (clear and coloured) with plastic screw caps. The bottles ranged from 500 mL to 750 mL volumes and were purchased in shops throughout the months of April, May and June 2021. The samples were kept under cool conditions and transported to the laboratory and stored at 4 °C until the chemical analysis. Water samples comprising three samples from the same brand were mixed together and poured into clean beakers before they were analysed for various parameters. Physicochemical assessments analysed according to the methods described in the American Public Health Association [36] and Gautum [3]. Analysis of variance (ANOVA) was performed to check if the differences obtained in the parameters from the different brands were significant.

#### Laboratory Analysis of the Water Samples

Physicochemical properties such as pH, electrical conductivity (EC), temperature and total dissolved solids (TDS) were measured using a multi-parameter analyser. Parameters such as nitrate, phosphate and sulphate concentrations were determined using a Shimadzu UV-1280 UV-VIS spectrophotometer. For each water sample, 5 mL was transferred into a cuvette cube and reagent was added to the cube, which was then placed inside the spectrophotometer for analysis.

### 2.2. Trace Element Analysis

Trace element analyses were performed using an inductively coupled plasma mass spectrometry (ICP-MS) flow injection analyser. The samples were digested by adding 3 mL of nitric acid to 10 mL of each of the water samples and allowing these to heat up for about 30 min. The resulting solutions were transferred into volumetric flasks and the volume made up by adding distilled water. The quality was assured by using reagents and chemicals of high purity and analytical grade. A multi-element solution from Perkin Elmer was used along with analytical grade (supra-pure) HNO_3_. The glassware and bottles used for the analysis were thoroughly cleaned with deionized water. The blank solution was repeated more than five times to ensure accuracy and that standard mean errors were kept to the barest minimum. The percentage recovery rates that was used were between 96 and 97% for all the trace metals. The quality assurance was further enhanced by spiking and homogenizing three replicates of five samples selected at random, and the recovery rates were all between 96 and 100% for all the trace elements.

#### Risk Assessment

A modified equation was used to determine the daily exposure for ingestion calculated according to the Dippong et al. [37] method:(1)$$\mathrm{CDI}=\frac{C\times \mathrm{DI}}{\mathrm{BW}}$$
where CDI stands for chronic intake (µg/kg/d), C stands for metal content in drinking water (µg/L), DI stands for average daily intake rate (L/d) and BW stands for body weight in kilograms. A second equation was used to calculate the hazard quotient for non-carcinogenic risk:(2)$$\mathrm{HQ}=\frac{\mathrm{CDI}}{\mathrm{RFD}}$$
where hazard quotient is HQ and the reference dosage is RFD (µg/kg/d). The SPSS program version 13 was used to perform descriptive statistics and one-way analysis of variance (ANOVA).

## 3. Results

Table 1 shows the physicochemical properties of the bottled water used in the study. From the results obtained, the values recorded for the pH for some of the bottled water were below the acceptable levels for drinking water. The water samples all showed in their label a pH between 6.5 and 8.5. However, the results obtained showed that samples 3, 6 and 9 were all within the acidic range with values ranging from 4.65 to 4.79. Generally, there were variations in the levels of pH from all the bottled water samples used in the study. Differences in the levels of pH for these bottled waters may suggest differences in the sources of the water. The study from Aris et al. [38] showed that the differences obtained might be due to the levels of calcium from the different sources which may raise the pH of the bottled water. Chiarenzelli and Pominville [39] have previously suggested that the geographical condition of the natural mineral-water sources could also contribute to the levels of the pH and that the dissolution that occurred in the basin may also affect the hydrogen-ion concentration. Our findings are similar to those reported by Ondieki et al. [40] where it was noted that the low pH observed in that study might have been due to soil geochemistry and other parameters such as acid rain in conjunction with SO_2_ and NOs. Some reports from literature have suggested that changes in water pH may affect the gut microbiota composition and the host metabolism while others have suggested an increase in the incidence of diabetes, especially from drinking water with low pH [41,42]. In addition, at low pH, the solubility of trace metals is enhanced which may increase their presence in the drinking water and this may have been the case in our study.

The levels of electrical conductivity (EC) and total dissolved solutes (TDS) were all within the acceptable range provided by the WHO. The EC ranged from 3 to 347 and the TDS from 2 to 171.67. The concentrations of nitrate and phosphate were all within the acceptable limit and in some instances below the detection limit (Table 1). The results of the electrical conductivity and the total dissolve solids were similar to the findings of Onidieki et al. [40] where the values obtained for EC ranged between 73.9 and 500 and were reported to be below the maximum guideline limit of 1500. It should be noted that EC determines the ability of the water to conduct electricity and is a function of the levels of Ca and Mg in water. The TDS on the other hand, determines the taste of the water if the concentrations are above the recommended limit of 1500 mg/L [13].

Table 2 shows the concentrations of the twelve trace metals studied from various bottled water samples of different brand names. The values in bold represent the values above the maximum allowable limit (MAL) set by the World Health Organization [43,44]. For instance, the concentrations obtained for Cr in this study were all above the WHO-recommended limit of 50 ug/L. The concentrations of Cr in this study were 4 or 5 times above the limit. Reports from other studies have suggested that the presence of Cr in drinking water may either be from natural or anthropogenic sources [45]. It could be suggested that some of the bottled water used in this study was a product of a natural source from borehole water as that is usually the practice, especially with unregistered stores. The soils from the area where these bottled waters were purchased is known to be rich in Cr and that might have accounted for the high concentrations of Cr in some of these bottled water [46]. The study of Roje and Sutalo [47] also suggested the need to purify the water before drinking because the baseline state of natural water might have been altered and it is therefore important to assess its usability for drinking. High concentrations of Cr in drinking water may pose a serious health risk to the individual relying on it. For instance, the study of Zhang and Li [48] and as reported by Zhitkovich [45] showed that there was an increased mortality due to stomach cancer among rural populations from the Liaoning Province of China where drinking water was highly polluted with Cr(VI) released by the ore-smelting facility.

The other element of concern with values higher than the WHO-recommended limit in our study is Pb. The concentrations of Pb from some of the water samples were above the WHO-recommended limit of 10 µg/L. Only three samples (5, 7 and 8) from our study had mean values below the acceptable limit; however, samples 5 and 8 with mean values above 9 µg/L may have had values closer to the acceptable limit owing to the standard deviation values obtained in this study. From literature, one of the major sources of Pb in drinking water is usually from the corrosion of pipes containing Pb. The findings of our study into the concentrations of Pb in drinking water are similar to those reported by Redmond et al. [49]. In all these studies, sources of Pb in the drinking water were linked to the corrosion of pipes and Kim et al. [50] further mentioned that environmental factors and the nature of the soil (whether it is polluted or not) may also affect the levels of Pb especially in underground water. Elevated blood Pb levels have been reported from individuals drinking Pb-contaminated water [51]. The main problem with Pb toxicity is the effects on the nervous system, both in children and adults, and Pb exposure during pregnancy has been linked with gestational pregnancy, miscarriage and premature death [52,53].

The values obtained for Ni from this study were all above the WHO-recommended limit of 20 µg/L. The concentrations of Ni in this study ranged from 74.79 ± 9.83 µg/L to 89.08 ± 0.65 µg/L. Ni is known as one of the heavy metals that may pose a serious health risk to humans [47]. Ni contamination of water, especially from the underground sources, may arise as a result of prolonged and direct exposure to minerals of underground water. With time, the Ni from the underground soil may mix with the water thereby increasing its concentration in the underground water. Ni may also enter the environment via fossil fuels in power plants, mines and refineries [54].

The other trace elements examined in this study, Fe, Cu, Mn, V, Ti, Mo and Cd were all below the WHO-recommended limit. It should be noted that in this study that the Pearson correlation showed a positive relationship between Ni and Cr (0.89) and Cr and Pb (0.21), although not a very strong correlation. These positive correlations, especially the correlation between Ni and Cr, may suggest a common source for these elements. Literature has suggested mining activities, among others, as one of the major sources of these elements in the environment [46,54]. The findings of the current study concur with those of the 2019 studies by Oladele and Digun-Aweto [55] who reported a strong correlation between nickel and chromium and connected this relationship to common sources such as industrial effluents, municipal wastes and soil-formation processes.

## 4. Human Health Risk Assessment

The human health risks from heavy metals was evaluated based on the proportion of the metals that were at hazardous levels and their presence in the water composition (Table 3).

The risk and quality assessment of trace metals for chronic daily intake (CDI) and hazard quotient (HQ) (Table 3) provided information on the possible health risks associated with drinking the bottled water used in this study. The minimum and maximum values for each trace metal were used to calculate the possible health risk. CDI values less than 10-4 showed no risk while values above this level represented a considerable health risk. From this study, the HQ index recorded for both the minimum and maximum levels for Cr and Pb posed a serious health risk, as noted in Table 3. The concentrations recorded for both trace metals from the study were all above the recorded safe limit for human consumption and might have accounted for the values obtained for the HQ. Similarly, the highest value obtained for Mn may also pose a health risk for those drinking the water. From Table 2 and Table 3, it was evident that samples 6, 9 and 12 may be considered unsafe for human consumption due to the levels of toxic trace metals present in the samples.

## 5. Conclusions

The current study examined the levels of trace metals in bottled water purchased from various retail stores, both registered and unregistered, in Pretoria, South Africa. The study showed, among other findings, that the levels of the hydrogen ions found in some of the bottled water were above the acceptable limit and within the acidic region which may facilitate the solubility of trace metals from the underground soils to the water. The values obtained for some of the toxic trace metals such as Pb, Ni and Cr examined in the study were clearly above the recommended limit safe for human consumption as set by WHO. It should be noted that South Africa has rich natural resources and this might have accounted for the high levels of these toxic trace metals in soil. The anthropogenic activities and improper waste disposal may also be a factor in the presence of these toxic trace metals in soil. The HQ calculated in the study showed that the continuous consumption of some of the bottled water used in the study may pose a serious health risk to consumers. The information provided on the attached label of some of the water samples did not correspond to the values obtained in our study. An effective monitoring program should be introduced that will assist with WHO compliance. This study concluded that compliance with the standard should be enforced and this may include an analytical report from the water supplier in order to inform users about the safety of the bottled water. Information on the levels of trace elements in the bottled water may be required due to the fact that South Africa has rich natural resources and the presence of trace elements in the soil might increase their levels in the drinking water.

## Figures and Tables

**Table 1 ijerph-19-15131-t001:** Physicochemical properties of studied bottled water samples.

Sample	TDS (mg/L)	EC (mS/m)	SO_4_ (mg/L)	NO_3_ (mg/L)	PO_4_ (mg/L)	pH
1	24.67	47	<0.05	0.24	<0.10	6.38
2	171.67	347.67	0.06	<0.20	<0.10	7.26
3	15.33	30.33	<0.05	0.22	<0.10	**4.67**
4	93.33	180.33	<0.05	<0.20	<0.10	6.68
5	26	50	<0.05	0.33	<0.10	5.33
6	71	145.67	<0.05	<0.20	<0.10	**4.77**
7	8.67	16.67	<0.05	<0.20	<0.10	5.93
8	2	3	<0.05	0.34	<0.10	5.59
9	68.67	137	<0.05	<0.20	<0.10	**4.75**
10	6.67	12.67	<0.05	0.31	<0.10	6.37
11	49.33	99.67	<0.05	0.34	<0.10	6.95
12	76.33	152.67	<0.05	0.1	<0.10	7.5
*MAL (WHO, 2006)	300	400	500	50	0.05	6.5

*MAL: maximum allowable limit. Bolded values are values not within the recommended limit.

**Table 2 ijerph-19-15131-t002:** The concentrations of trace elements (ug/L) in the bottled water samples.

Sample ID	Trace Elements											
	Ti	V	Cr	Mn	Fe	Ni	Cu	Zn	As	Mo	Cd	Pb
1	8.42 ± 1.12	1.45 ± 1.11	147.07 ± 1.23	72.21 ± 1.11	1245.35 ± 2.98	79.20 ± 1.55	149.93 ± 0.98	47.75 ± 2.22	1.25 ± 0.11	5.11 ± 0.09	0.18 ± 0.02	20.70 ± 1.10
2	4.94 ± 2.31	0.74 ± 0.89	142.53 ± 2.32	51.30 ± 0.76	884.56 ± 0.88	74.79 ± 9.83	137.97 ± 1.45	40.74 ± 1.34	1.93 ± 0.87	5.31 ± 0.10	0.18 ± 0.03	16.10 ± 0.98
3	4.50 ± 1.11	0.97 ± 0.05	154.38 ± 1.11	129.22 ± 2.02	1053.21 ± 2.42	79.98 ± 0.55	182.86 ± 3.22	46.39 ± 5.34	1.31 ± 0.12	5.18 ± 0.21	0.16 ± 0.01	14.11 ± 0.88
4	13.16 ± 0.87	6.61 ± 1.02	151.38 ± 4.36	128.17 ± 1.31	1346.91 ± 3.77	80.90 ± 2.21	137.87 ± 0.77	2978.31 ± 9.76	1.02 ± 0.06	5.08 ± 0.12	0.16 ± 0.01	12.23 ± 1.22
5	5.68 ± 0.24	0.73 ± 0.06	149.31 ± 0.89	102.71 ± 2.43	1048.77 ± 0.88	79.82 ± 0.87	171.96 ± 1.99	51.20 ± 1.09	1.35 ± 0.01	4.98 ± 0.04	0.17 ± 0.02	9.43 ± 0.98
6	6.80± 1.91	1.08 ± 0.23	147.43 ± 1.43	231.80 ± 0.83	1153.08 ± 2.54	80.14 ± 3.11	200.68 ± 2.65	55.35 ± 7.11	0.92 ± 0.03	5.17 ± 0.01	0.18 ± 0.09	123.15 ± 1.12
7	6.51 ± 0.88	1.33 ± 0.21	161.61 ± 2.11	63.61 ± 1.43	927.88 ± 0.98	83.70 ± 0.88	177.37 ± 7.11	40.92 ± 2.09	1.18 ± 0.23	5.27 ± 0.22	0.17 ± 0.10	9.21 ± 0.11
8	14.81 ± 0.65	2.19 ± 0.65	153.91 ± 4.34	16.95 ± 0.77	1144.88 ± 3.45	83.41 ± 0.11	21.33 ± 0.84	11.30 ± 0.65	0.65 ± 0.05	4.80 ± 0.89	0.16 ± 0.08	5.98 ± 0.43
9	6.61 ± 1.12	1.25 ± 0.43	156.38 ± 1.87	268.91 ± 2.65	1241.53 ± 2.35	83.42 ± 2.45	184.91 ± 3.20	48.37 ± 0.11	1.52 ± 0.11	5.51 ± 0.32	0.18 ± 0.04	12.73 ± 0.61
10	4.85 ± 0.98	0.94 ± 0.11	156.03 ± 9.08	102.01 ± 0.77	1087.36 ± 7.11	81.93 ± 3.01	228.11 ± 7.54	62.12 ± 0.34	1.47 ± 0.06	5.38 ± 1.22	0.17 ± 0.07	25.58 ± 2.03
11	5.25 ± 1.03	1.87 ± 0.43	164.70 ± 4.03	63.09 ± 3.67	942.34 ± 5.45	86.35 ± 9.02	206.79 ± 3.25	49.86 ± 0.54	1.68 ± 0.09	5.61 ± 0.89	0.15 ± 0.01	78.56 ± 1.98
12	9.97 ± 2.11	5.03 ± 0.98	190.26 ± 2.09	93.49 ± 2.98	895.74 ± 0.77	89.08 ± 0.65	407.32 ± 1.01	136.37 ± 0.33	1.96 ± 0.02	5.45 ± 1.01	0.67 ± 0.09	57.68 ± 0.31

**Table 3 ijerph-19-15131-t003:** Risk indices for daily intake and hazard quotient.

	Chronic Daily Intake (mg/kg)		Hazard Quotient (HQ)	
Metals	Minimum	Maximum	Minimum	Maximum
Ti	3.1 × 10^−2^	9.8 × 10^−5^	1 × 10^−2^	3.2 × 10^−5^
V	5.1 × 10^−6^	4.6 × 10^−5^	7.3 × 10^−4^	6.6 × 10^−3^
Cr	9.9 × 10^−4^	1.3 × 10^−3^	0.33	0.43
Mn	1.2 × 10^−4^	1.9 × 10^−3^	8.6 × 10^−3^	0.14
Fe	6.1 × 10^−3^	9.3 × 10^−3^	8.7 × 10^−3^	1.3 × 10^−2^
Ni	5.2 × 10^−4^	6.2 × 10^−4^	2.6 × 10^−2^	3.1 × 10^−2^
Cu	1.5 × 10 ^−4^	2.8 × 10^−3^	3.8 × 10^−3^	7 × 10^−2^
Zn	7.8 × 10^−5^	2 × 10^−2^	2.6 × 10^−4^	6.6 × 10^−2^
As	4.5 × 10^−6^	1.4 × 10^−5^	1.5 × 10^−2^	4.6 × 10^−2^
Mo	3.3 × 10^−5^	3.9 × 10^−5^	6.6 × 10^−3^	7.8 × 10^−3^
Cd	1 × 10^−6^	4.6 × 10^−6^	9.2 × 10^−4^	8.8 × 10^−3^
Pb	4.1 × 10^−5^	8.5 × 10^−4^	1.2 × 10^−2^	0.2

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
