# Peer review of "Health Risk Assessment of Trace Metals in Bottled Water Purchased from Various Retail Stores in Pretoria, South Africa"

_ijerph, 2022, doi:10.3390/ijerph192215131_

Round 1

Reviewer 1 Report

This is very straightforward study on bottled water. The results are alarming - in particular the highly elevated Cr, Ni, Pb. Overall, the work fits the scope of the journal and more important, the findings will be of great interest to the residents of Pretoria and South Africa as a whole (assuming the studied bottled water brands are sold all over the country). Few minor remarks below:

Overall the paper is understandable but the English needs improvement;

Line 127 - why do you need to “homogenize” water, it is already homogeneous in the bottle??

Blank – show the results for the blank measurements;

What standards were used for quantification of the ICP-MS results? Any known standard analyzed with the samples?

Table 1 – define “MAL” in the Table caption (yes, it is defined later on in the text but you need to define it here first as not everyone is aware of this abbreviation);

Table 2 – list units, again you can do that in the Table captions – e.g., all concentrations are ppb or micg/L..

Table 2 – add MAL for the metals here too, same as Table 1.

Line 226 – rewrite this sentence as this is not accurate. Not all “were above the WHO limit” - 3 are below (or close) to the limit as mentioned below.

Line 246 – “stones” is not OK term, use minerals (or rocks) instead.

Lines 282-283 – rewrite as this is the other way around. Low pH = increase in H+ so “the hydrogen ions” will be above, not below the limit (but the pH will be below the limit);

Author Response

Dear Reviewer,

Thanks so much for your inouts and highly appreciated.

We have answered all your queries and hopefully this is good. Please do not hesitate to contac us if there are other queries.

Thanks

Prof Olowoyo JO

Reviewer 2 Report

Manuscript ID: ijerph-2005205
Title: Health risk assessment of trace metals in bottled water purchased from various retail stores in Pretoria, South Africa.
OVERVIEW
The study investigates the concentrations of trace metals and physico -chemical properties of bottled water purchased from various supermarkets (registered and unregistered) in Pretoria to determine the health risk that may be associated with the levels of trace metals resulting from the consumption of bottled water.
GENERAL COMMENTS
The subject matter is actual, interesting and within the scope of the IJERPH.
The manuscript complies with the journal template and is well structured.
The title describes the manuscript and is appropriate.
The English spelling and grammar are fine.
My major concern is that the number of samples (twelve samples from twelve different brands?) is rather small and one bottle from each source may not represent all the commercialized bottles from that source.
 As for the rest, I have some suggestions. Please read the specific comments.
In conclusion, I believe this manuscript is interesting and worthy of publication after major changes.
SPECIFIC COMMENTS
Please consider increasing the number of samples from each water source to the values from the physico-chemical variables to become statistically representative of the commercially available bottles.
If possible, also report the values of the physical-chemical parameters printed on the label of the bottles, in the official reports and from other scientific studies (if available).

Author Response

Dear Reviewer

Thanks so much for your inputs.

We did not have a copy of what you corrected but your major concern in form of a note was answered.

Please note that we did not just buy 12 samples but about 48 samples and were mixed as a representative sample. We have attempted to read the manuscript again and correct some typographical errors. Thanks so much for your input.

Round 2

Reviewer 2 Report

Manuscript ID: ijerph-2005205 (revised version review)
Title: Health risk assessment of trace metals in bottled water purchased from various retail stores in Pretoria, South Africa.

GENERAL COMMENTS
The authors have improved the manuscript and answered all the questions posted in the first review.
In my opinion, the manuscript may be published as it is.